# Shannon Entropy Estimation for Linear Processes

**Timothy Fortune [1] and Hailin Sang [2,*]** 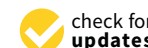

[1]   Department of Statistics, University of Connecticut, Storrs, CT 06269, USA; timothy.fortune@uconn.edu
[2]   Department of Mathematics, University of Mississippi, University, MS 38677, USA
[*]   Correspondence: sang@olemiss.edu

**Abstract:** In this paper, we estimate the Shannon entropy $S(f) = -\mathbb{E}\left[\log(f(x))\right]$ of a one-sided linear process with probability density function $f(x)$. We employ the integral estimator $S_n(f)$, which utilizes the standard kernel density estimator $f_n(x)$ of $f(x)$. We show that $S_n(f)$ converges to $S(f)$ almost surely and in $\text{Ł}^2$ under reasonable conditions.

**Keywords:** linear process; kernel entropy estimation; Shannon entropy

## 1. Introduction

Let $f(x)$ be the common probability density function of a sequence $\{X_n\}_{n=1}^{\infty}$ of identically distributed observations. The associated Shannon entropy

$$S(f) = \mathbb{E}\left[-\log f(X)\right] = -\int f(x)\,\log f(x)\,dx \tag{1}$$

of such an observation was first introduced by Shannon (1948). In his 1948 paper, Shannon utilized this tool in his mathematical investigation of the theory of communication. Today, entropy is widely applied in the fields of information theory, statistical classification, pattern recognition and so on, since it is a measure of the amount of uncertainty present in a probability distribution.

In the literature, several estimators for the Shannon entropy have been introduced. See Beirlant et al. (1997) for an overview. Many of these estimators have been studied in cases where the data are independent. In 1976, Ahmad and Lin (1976) obtained results using the resubstitution estimator $H_n = -\frac{1}{n}\sum_{i=1}^{n}\ln f_n(X_i)$ for independent data $\{X_i\}_{i=1}^{n}$. In particular, he showed consistency in the first and second mean under certain regularity conditions. Here, $f_n(x)$ is the kernel density estimator. Dmitriev and Tarasenko (1973) reported results in 1973 for estimating functionals of the type $\int H\big(f(x), f'(x), \ldots, f^k(x)\big)dx$, where the common density $f(x)$ of the independent $X_i$ is assumed to have at least $k$ derivatives. Plugging in kernel density estimators (see their paper and references therein) for the arguments of $H$ and integrating only over the symmetric interval $[-k_n, k_n]$, which is determined by a sequence $\{k_n\}_{n=1}^{\infty}$ of a certain order, they provided a result for the estimation of Shannon entropy using the estimator that Beirlant et al. (1997) refer to as the integral estimator. Their results give conditions for almost sure convergence.

Interestingly enough, Dmitriev and Tarasenko (1973) also provided (because their work is a more general investigation of functionals) a result for the estimation of the quadratic Rényi entropy $Q(f) = \int f^2(x)dx$. Conditions are provided specifically for the almost sure convergence of their estimator to the true value $Q(f)$. The estimation of Rényi entropy for the dependent case is challenging. A dependent case is treated by Sang et al. (2018). They studied the estimation of the quadratic Entropy for the one-sided linear process. Utilizing the Fourier transform along with the projection method, they demonstrate that the kernel entropy estimator satisfies a central limit theorem for short memory linear processes.

To study the Shannon entropy for dependent data is also a challenging problem, and to the best of our knowledge, general results for the Shannon entropy estimation of regular time series data are still unknown. In this paper, we study the Shannon entropy $S(f)$ for the one-sided linear process

$$X_n = \sum_{i=0}^{\infty} a_i \varepsilon_{n-i}, \tag{2}$$

where the innovations $\varepsilon_i$ are independent and identically distributed real valued random variables on some probability space $(\Omega, \mathcal{F}, \mathbb{P})$ with mean zero and finite variance $\sigma_\varepsilon^2$ and where the collection $\{a_i : i \geq 0\}$ of real coefficients satisfies $\sum_{i=0}^{\infty} a_i^2 < \infty$. Additionally, we will require that the common density $f_\varepsilon(x)$ of the innovations be bounded. The estimator we utilize employs the kernel method, which was first introduced by Rosenblatt (1956); Parzen (1962). The kernel estimator will be denoted by

$$f_n(x) = \frac{1}{nh_n} \sum_{i=1}^{n} K\left(\frac{x - X_i}{h_n}\right), \tag{3}$$

where the sequence $\{h_n\}_{n=1}^{\infty}$ provides the bandwidths, and $K : \mathbb{R} \to \mathbb{R}$ is the kernel function which satisfies $\int_{\mathbb{R}} K(x)dx = 1$. Typically, the kernel function is a probability density function.

This method has proven to be successful in estimating probability density functions and their derivatives, regression functions, etc., in both the independent and dependent setting. For the independent setting, see the books (Devroye and Györfi (1985); Silverman (1986); Nadaraya (1989); Wand and Jones (1995); Schimek (2000); Scott (2015)) and the references therein. For the dependent setting, we refer the reader to (Tran (1992); Honda (2000); Wu and Mielniczuk (2002); Wu et al. (2010)). Bandwidth selection is an important issue in kernel density estimation, and there is a lot of research in this direction. See, e.g., Duin (1976); Rudemo (1982); Slaoui (2014 2018).

A few remarks about notation and terms used in the paper follow. Let $\{a_n\}_{n=1}^{\infty}$ and $\{b_n\}_{n=1}^{\infty}$ be real-valued sequences. By $a_n = o(b_n)$ we understand that $a_n/b_n \to 0$ and $a_n = O(b_n)$ means that $\limsup |a_n/b_n| < C$ for some positive number $C$. Essentially, this is the standard Landau little oh and big oh notation. When we write, $a_n \ll b_n$, we mean $a_n = o(b_n)$, and as one might guess, $b_n \gg a_n$ means $a_n \ll b_n$. We also employ the notation $a_n \asymp b_n$ to indicate that $0 < \liminf_{n\to\infty} \frac{a_n}{b_n} \leq \limsup_{n\to\infty} \frac{a_n}{b_n} < \infty$. A function $l : [0, \infty) \to \mathbb{R}$ is referred to as *slowly varying* (at $\infty$) if it is positive and measurable on $[A, \infty)$ for some $A \in \mathbb{R}^+$ such that $\lim_{x\to\infty} l(\lambda x)/l(x) = 1$ holds for each $\lambda \in \mathbb{R}^+$. The set of all functions $g : \mathbb{R} \to \mathbb{R}$ which are *Hölder continuous* of some order $r$ will be denoted as $\mathcal{C}^r(\mathbb{R})$. That is, for each $g \in \mathcal{C}^r(\mathbb{R})$ there exists $C_g \in \mathbb{R}^+$, such that for all $x, x' \in \mathbb{R}$, we have $|g(x) - g(x')| \leq C_g|x - x'|^r$, and when $r = 1$, we recognize this as the well-known *Lipschitz condition*. The notation $Ł^p(E)$ with $0 < p < \infty$ represents the set of all real-valued functions $f$ defined on some measure space $(E, \mathcal{A}, \mu)$ having the property that $\int_E |f(x)|^p \, d\mu < \infty$. In the case that $E = \mathbb{R}$ and unless otherwise specified, the measure $\mu$ is tacitly understood to be Lebesgue measure and $\mathcal{A}$ is assumed to contain the Borel sets. $Ł^\infty(E)$ refers to the set of real-valued functions defined on $E$ which are bounded almost everywhere. Whenever the domain space of the function is understood, we may simply write $Ł^p$.

The following are *bandwidth, kernel, and density conditions* that we shall refer to throughout this paper:

**B.1** $h_n \asymp (n^{-1} \log n)^{\frac{1}{5}}$;
**K.1** $K \in \mathcal{C}^\iota(\mathbb{R})$ for some $\iota \in (0, 1]$ is bounded with bounded support;
**K.2** $\int uK(u) \, du = 0$;
**D.1** $f_\varepsilon, f_\varepsilon', f_\varepsilon'' \in Ł^\infty(\mathbb{R})$;
**D.2** $f_\varepsilon, f_\varepsilon', f_\varepsilon'' \in Ł^2(\mathbb{R})$;
**D.3** $f'' \in Ł^\infty(\mathbb{R})$.

Notice that the bandwidth, kernel, and density conditions are prefixed using **B**, **K**, and **D**, respectively.

In this first section, we have provided an introduction to the problem, a survey of past research in this area, and the notation to be used throughout. The main results are reported in Section Two. In Section Three, we present the proofs of the main results. Finally, the Appendix A introduces the reader to foundational results, which will be required in the proof of our main results.

## 2. Main Results

If $\{\varepsilon_i : i \in \mathbb{Z}\}$ is a sequence of independent and identically distributed random variables over a common probability space $(\Omega, \mathcal{F}, \mathbb{P})$ in $\mathcal{L}^q(\Omega)$ for some $q > 0$, $\mathbb{E}\,\varepsilon_i = 0$ when $q \geq 1$, and $\{a_i\}_{i=0}^{\infty}$ is a sequence of real coefficients such that $\sum_{i=0}^{\infty} |a_i|^{2 \wedge q} < \infty$, then the linear process $X_n$ given in (2) exists and is well-defined. For the case $q \geq 2$ where the innovations have finite variance, we say that the process has short memory (short-range dependence) if $\sum_{i=0}^{\infty} |a_i| < \infty$ and $\sum_{i=0}^{\infty} a_i \neq 0$ and long memory (long-range dependence) otherwise. Throughout, we assume that each $\varepsilon_i \in \mathcal{L}^q$ with $q \geq 2$.

Let $f(x)$ be the probability density function of the linear process $X_n = \sum_{i=0}^{\infty} a_i \varepsilon_{n-i}$, $n \in \mathbb{N}$ defined in (2). In this paper, we estimate the Shannon Entropy $-\int f(x) \log f(x)\, dx$ of the linear process. To do this, we employ the integral estimator

$$S_n(f) = - \int_{A_n} f_n(x)\, \log f_n(x)\, dx, \tag{4}$$

where $f_n(x)$ is the standard kernel density estimator defined in (3). The (random) sets $A_n$ are given by

$$A_n = \{x \in \mathbb{R} : 0 < \gamma_n \leq f_n(x)\}, \tag{5}$$

where $\{\gamma_n\}_{n=1}^{\infty}$ is an appropriately defined sequence in $\mathbb{R}^+$ that converges to zero.

Our estimator utilizes the kernel method of density estimation, and we will accordingly require adherence of the kernel to certain conditions. In addition, we impose some conditions on the bandwidths and on some of the densities of the problem. These conditions were listed in the previous section. Based on these conditions, let us consider the properties of the estimator (4). We proceed in a manner similar to the analysis done by Bouzebda and Elhattab (2011) for the independent case.

**Theorem 1.** *Let $\{X_n : n \in \mathbb{N}\}$ be the linear process given in (2), and assume that it has short memory. Furthermore, assume that $S(f)$ is finite. If the bandwidth, kernel, and density conditions listed earlier are satisfied, then*

$$\limsup_{n \to \infty} \left( \frac{n\,\gamma_n^5}{\log n} \right)^{\frac{2}{5}} \left| S_n(f) - \int_{A_n} \left( -\log f(x) \right) f(x)\, dx \right|$$

*is bounded almost surely whenever the condition $\gamma_n \gg h_n$ is imposed on the sequence $\{\gamma_n\}_{n=1}^{\infty}$.*

**Corollary 1.** *If the conditions of Theorem 1 hold, then we have*

$$\lim_{n \to \infty} |S_n(f) - S(f)| = 0$$

*almost surely.*

**Theorem 2.** *Let $\{X_n : n \in \mathbb{N}\}$ be the linear process given in (2), and assume that it has short memory. Furthermore, assume that $S(f)$ is finite. If the bandwidth, kernel, and density conditions listed earlier are satisfied, then*

$$\limsup_{n \to \infty} \left( \frac{n \, \gamma_n^5}{\log n} \right)^{\frac{2}{5}} \left\| S_n(f) - \int_{A_n} \left( -\log f(x) \right) f(x) \, dx \right\|_2 \tag{6}$$

*is bounded whenever the condition $\gamma_n \gg h_n$ is imposed on the sequence $\{\gamma_n\}_{n=1}^{\infty}$.*

**Corollary 2.** *If the conditions of Theorem 2 hold, then the mean squared error (MSE) satisfies*

$$\lim_{n \to \infty} \mathrm{MSE}(S_n(f)) = 0.$$

**Remark 1.** *In this paper, we work on the entropy estimation for short memory linear processes by applying the integral method. It is interesting to know whether the similar results hold for long memory linear processes. It is also interesting to know whether the resubstitution method works for dependent data such as linear processes. However, the research in these directions are beyond the scope of this paper. We leave research in these directions for future work.*

**Remark 2.** *In a wide range of disciplines, including finance, geology, and engineering, many time series may be modeled using a linear process. In such instances, our result provides a method for estimating the associated Shannon Entropy. One example is the discriminatory data on the arrival phases of earthquakes and explosions, which were captured at a seismic recording station. Another example is the data about returns on the New York Stock Exchange. See these and many other time series data in the book by Shumway and Stoffer (2011) and other books on time series.*

## 3. Proofs

**Lemma 1.** *If the conditions of Theorem 1 (or Theorem 2) hold, then*

$$\sup_{x \in \mathbb{R}} |f_n(x) - f(x)| = O\left( \left( \frac{\log n}{n} \right)^{\frac{2}{5}} \right) \tag{7}$$

*almost surely.*

**Proof.** This lemma follows from Theorem 2 of Wu et al. (2010) (see their discussion immediately after the statement of Theorem 2 and in the penultimate paragraph of section 4.1). See also the discussion in the Appendix A on fundamental results. □

**Lemma 2.** *If the conditions of Theorem 1 (or Theorem 2) hold, then*

$$\gamma_n^5 \gg \frac{\log n}{n}. \tag{8}$$

**Proof.** Because $h_n \asymp (n^{-1} \log n)^{\frac{1}{5}}$, there exists $C \in \mathbb{R}^+$ such that

$$\frac{h_n^5}{n^{-1} \log n} > C$$

for sufficiently large $n$. Therefore,

$$\lim_{n\to\infty} \frac{\gamma_n^5}{n^{-1}\log n} = \lim_{n\to\infty} \frac{\gamma_n^5}{h_n^5} \cdot \frac{h_n^5}{n^{-1}\log n}$$

$$\geq C \lim_{n\to\infty} \left(\frac{\gamma_n}{h_n}\right)^5 \to \infty$$

as $n \to \infty$, from which (8) follows. $\square$

**Note.** Our use of Lemma 2 in the proofs of Theorems 1 and 2 will be tacit.

**Lemma 3.** *If $\nu$ is a finite signed measure that is absolutely continuous with respect to a measure $\mu$, then corresponding to every positive number $\varepsilon$ there is a positive number $\delta$ such that $|\nu|(E) < \varepsilon$ whenever $E$ is a measurable set for which $\mu(E) < \delta$.*

**Proof.** This is a basic result from measure theory. See, for example, Theorem B of Halmos (1974) in section 30. $\square$

**Proof of Theorem 1.** We begin with the decomposition

$$S_n(f) - \int_{A_n} \left(-\log f(x)\right) f(x)\, dx$$

$$= -\int_{A_n} f_n(x)\, \log f_n(x)\, dx + \int_{A_n} f(x)\, \log f(x)\, dx$$

$$= -\int_{A_n} f_n(x)\, \log f_n(x)\, dx + \int_{A_n} f(x)\, \log f_n(x)\, dx \tag{9}$$

$$- \int_{A_n} f(x)\, \log f_n(x)\, dx + \int_{A_n} f(x)\, \log f(x)\, dx$$

$$= I_{n,1} + I_{n,2},$$

where

$$I_{n,1} := -\int_{A_n} \left[f_n(x) - f(x)\right]\, \log f_n(x)\, dx,$$

and

$$I_{n,2} := -\int_{A_n} f(x) \left[\log f_n(x) - \log f(x)\right]\, dx.$$

First, we consider $I_{n,1}$. Using the inequality

$$|\log z| \leq z + \frac{1}{z}$$

for $z \in \mathbb{R}^+$, we notice that for all $x \in A_n$, we have

$$\left|\log f_n(x)\right| \leq f_n(x) + \frac{1}{f_n(x)}$$

$$= \left(1 + \frac{1}{(f_n(x))^2}\right) f_n(x)$$

$$\leq \left(1 + \frac{1}{\gamma_n^2}\right) f_n(x).$$

It follows that

$$\begin{aligned}
|I_{n,1}| &\leq \sup_{x \in \mathbb{R}} |f_n(x) - f(x)| \int_{A_n} |\log f_n(x)| \; dx \\
&\leq \left(1 + \frac{1}{\gamma_n^2}\right) \sup_{x \in \mathbb{R}} |f_n(x) - f(x)|,
\end{aligned}$$
(10)

since $f_n(x)$ integrates to unity over the real line.

Next, we consider $I_{n,2}$. Since the set over which we are integrating may be changed to $A_n \cap \{x : f(x) > 0\}$ without affecting the value of $I_{n,2}$, we may assume that $f$ is positive on $A_n$. Using the inequality

$$\log z \leq |z - 1| + |z^{-1} - 1|$$

for $z \in \mathbb{R}^+$, we notice that for all $x \in A_n$, we have

$$\begin{aligned}
\left|\log f_n(x) - \log f(x)\right| &= \left|\ln\left(\frac{f_n(x)}{f(x)}\right)\right| \\
&\leq \left|\frac{f_n(x)}{f(x)} - 1\right| + \left|\frac{f(x)}{f_n(x)} - 1\right| \\
&= \left|\frac{f_n(x) - f(x)}{f(x)}\right| + \left|\frac{f(x) - f_n(x)}{f_n(x)}\right| \\
&= \left(1 + \frac{f_n(x)}{f(x)}\right)\left|\frac{f_n(x) - f(x)}{f_n(x)}\right| \\
&\leq \frac{C}{\gamma_n}|f_n(x) - f(x)|,
\end{aligned}$$
(11)

if we can justify the existence of $C \in \mathbb{R}^+$. To that end, define

$$\varepsilon_n = \sup_{x \in A_n} |f_n(x) - f(x)|,$$

and note that for all $x \in A_n$, we have

$$\left|1 - \frac{f(x)}{f_n(x)}\right| \leq \frac{\varepsilon_n}{f_n(x)} \leq \frac{\varepsilon_n}{\gamma_n}.$$

Taking the supremem over $A_n$ yields

$$\sup_{x \in A_n} \left|1 - \frac{f(x)}{f_n(x)}\right| \leq \frac{\varepsilon_n}{\gamma_n} = \gamma_n^{-1} \varepsilon_n$$

$$\leq C\gamma_n^{-1} \Big/ \left(\frac{n}{\log n}\right)^{\frac{2}{5}},$$

by Lemma 1. Note that

$$\gamma_n^{-1} = o\left(\left(\frac{n}{\log n}\right)^{\frac{2}{5}}\right),$$

since

$$\lim_{n\to\infty} \frac{\gamma_n^{-1}}{\left(\frac{n}{\log n}\right)^{\frac{2}{5}}} = \lim_{n\to\infty} \frac{\left(\frac{\log n}{n}\right)^{\frac{2}{5}}}{\gamma_n}$$

$$= \lim_{n\to\infty} \left( \frac{\frac{\log n}{n}}{\gamma_n^5} \frac{\frac{\log n}{n}}{1} \right)^{\frac{1}{5}}$$

$$= 0.$$

This guarantees the existence we sought to establish. We continue with

$$\left| I_{n,2} \right| \le \frac{C}{\gamma_n} \sup_{x\in\mathbb{R}} \left| f_n(x) - f(x) \right| \int_{A_n} f(x)\, dx$$

$$\le \frac{C}{\gamma_n} \sup_{x\in\mathbb{R}} \left| f_n(x) - f(x) \right|, \tag{12}$$

since $f_n(x)$ integrates to unity over the real line.

In view of (9), (10) and (12), we have shown that

$$\left| S_n(f) - \int_{A_n} \left( -\log f(x) \right) f(x)\, dx \right| \tag{13}$$

$$\le \left( \frac{1}{\gamma_n^2} + \frac{C}{\gamma_n} + 1 \right) \sup_{x\in\mathbb{R}} \left| f_n(x) - f(x) \right|. \tag{14}$$

Therefore,

$$\limsup_{n\to\infty} \left( \frac{n\,\gamma_n^5}{\log n} \right)^{\frac{2}{5}} \left| S_n(f) - \int_{A_n} \left( -\log f(x) \right) f(x)\, dx \right|$$

$$\le \limsup_{n\to\infty} \left( \frac{n}{\log n} \right)^{\frac{2}{5}} \gamma_n^2 \left( \frac{1}{\gamma_n^2} + \frac{C}{\gamma_n} + 1 \right) \sup_{x\in\mathbb{R}} \left| f_n(x) - f(x) \right|$$

$$= \limsup_{n\to\infty} \left( \gamma_n^2 + C\gamma_n + 1 \right) \left( \frac{n}{\log n} \right)^{\frac{2}{5}} \sup_{x\in\mathbb{R}} \left| f_n(x) - f(x) \right|,$$

where the last expression is constant almost surely by Lemma 1 and since $\gamma_n \to 0$. ☐

**Proof of Corollary 1.** By the triangle inequality

$$\left| S_n(f) - S(f) \right| \le J_{n,1} + J_{n,2},$$

where

$$J_{n,1} = \left| S_n(f) - \int_{A_n} \left( -\log f(x) \right) f(x)\, dx \right|$$

and

$$J_{n,2} = \left| \int_{A_n} \left( -\log f(x) \right) f(x)\, dx - S(f) \right|.$$

Since $J_{n,1} \to 0$ almost surely by Theorem 1, we only need to contend with $J_{n,2}$. That is, we need to show that

$$\int_{A_n^c} f(x) \log f(x) \, dx \to 0 \tag{15}$$

almost surely as $n \to \infty$.

For any Borel measurable set $E$, consider

$$P(E) = \int_E f(x) \, dx,$$

and define the signed measure

$$\nu(E) = -\int_E \log f(x) \, dP.$$

Since $|S(f)| < \infty$, both $\nu^+$ and $\nu^-$ are finite measures, and thus, $\nu$ is a finite signed measure that is absolutely continuous with respect to $P$. Because of Lemma 3, it suffices for us to demonstrate that

$$P(A_n^c) \to 0$$

almost surely. For any $x \in A_n^c$, we have $f_n(x) < \gamma_n$. By Lemma 1, there exists $C \in \mathbb{R}^+$ such that $f(x) \leq f_n(x) + |f_n(x) - f(x)| < \gamma_n + C\left(\frac{\log n}{n}\right)^{\frac{2}{5}}$ almost surely, and hence, we have shown that $A_n^c \subseteq B_n$ almost surely, where

$$B_n := \left\{ x : f(x) < \gamma_n + C\left(\frac{\log n}{n}\right)^{\frac{2}{5}} \right\}.$$

It is easy to see that

$$0 \leq P(A_n^c) \leq P(B_n) \to 0$$

almost surely, since $\gamma_n + C\left(\frac{\log n}{n}\right)^{\frac{2}{5}} \to 0$ as $n \to \infty$. $\square$

**Proof of Theorem 2.** We start with

$$\left\| S_n(f) - \int_{A_n} \left( -\log f(x) \right) f(x) \, dx \right\|_2$$

$$\leq \left\| S_n(f) + \int_{A_n} f_n(x) \log f(x) \, dx \right\|_2$$

$$+ \left\| \int_{A_n} f_n(x) \log f(x) \, dx - \int_{A_n} f(x) \log f(x) \, dx \right\|_2$$

$$=: K_{n,1} + K_{n,2}.$$

Recall inequality (11) in the proof of Theorem 1. Arguing in a similar manner as before, we can demonstrate the existence of $C_1 \in \mathbb{R}^+$ so that

$$
\begin{aligned}
K_{n,1} &= \left\| \int_{A_n} f_n(x) \left[ \log f(x) - \log f_n(x) \right] dx \right\|_2 \\
&= \left\| \int_{A_n} f_n(x) \log \frac{f(x)}{f_n(x)} \, dx \right\|_2 \\
&\leq \left\| \int_{A_n} f_n(x) \left| \log \frac{f(x)}{f_n(x)} \right| dx \right\|_2 \\
&\leq \left\| \int_{A_n} \frac{C_1}{\gamma_n} |f_n(x) - f(x)| f_n(x) \, dx \right\|_2 \\
&\leq \frac{C_1}{\gamma_n} \left( \frac{\log n}{n} \right)^{\frac{2}{5}} \left\| \int_{x \in A_n} f_n(x) \, dx \right\|_2 \\
&\leq \frac{C_1}{\gamma_n} \left( \frac{\log n}{n} \right)^{\frac{2}{5}}.
\end{aligned}
$$

Notice also that

$$
\begin{aligned}
K_{n,2} &= \left\| \int_{A_n} [f_n(x) - f(x)] \log f(x) \, dx \right\|_2 \\
&\leq \left\| \int_{A_n} |f_n(x) - f(x)| \log f(x) \, dx \right\|_2 \\
&\leq C_2 \left( \frac{\log n}{n} \right)^{\frac{2}{5}} \left\| \int_{A_n} \left( f(x) + \frac{1}{f(x)} \right) dx \right\|_2 \\
&\leq C_2 \left( \frac{\log n}{n} \right)^{\frac{2}{5}} \left[ 1 + \left\| \int_{A_n} \frac{f_n(x)}{f(x)} \frac{1}{f_n^2(x)} f_n(x) \, dx \right\|_2 \right] \\
&\leq C_2 \left( \frac{\log n}{n} \right)^{\frac{2}{5}} \left( 1 + \frac{C_3}{\gamma_n^2} \right).
\end{aligned}
$$

Therefore,

$$
\begin{aligned}
\left( \frac{n \, \gamma_n^5}{\log n} \right)^{\frac{2}{5}} \left\| S_n(f) - \int_{A_n} \left( -\log f(x) \right) f(x) \, dx \right\|_2 \\
\leq \left( \frac{n}{\log n} \right)^{\frac{2}{5}} \gamma_n^2 \left[ \frac{C_1}{\gamma_n} \left( \frac{\log n}{n} \right)^{\frac{2}{5}} + C_2 \left( \frac{\log n}{n} \right)^{\frac{2}{5}} \left( 1 + \frac{C_3}{\gamma_n^2} \right) \right] \\
= C_1 \gamma_n + C_2 \gamma_n^2 + C_2 C_3,
\end{aligned}
$$

from which the result follows. □

**Proof of Corollary 2.** Note the decomposition

$$
\begin{aligned}
\sqrt{\mathrm{MSE}(S_n(f))} &= \left\| S_n(f) - S(f) \right\|_2 \\
&\leq \left\| S_n(f) - \int_{A_n} \left( -\log f(x) \right) f(x) \, dx \right\|_2 \\
&\quad + \left\| \int_{A_n} \left( -\log f(x) \right) f(x) \, dx - S(f) \right\|_2 \\
&=: M_{n,1} + M_{n,2}.
\end{aligned}
$$

By Theorem 2, $M_{n,1} \to 0$. Now, let

$$W_n = \int_{x \in \mathbb{R}} \int_{y \in \mathbb{R}} f(x)f(y) \log f(x) \log f(y) \, \mathcal{I}\big((x,y) \in A_n^c \times A_n^c\big) \, dx \, dy$$

and

$$W = \int_{x \in \mathbb{R}} \int_{y \in \mathbb{R}} f(x)f(y) \big| \log f(x) \log f(y) \big| \, dx \, dy.$$

Recall from (15) in the proof of Corollary 1 that $W_n \to 0$ almost surely. Because $|S(f)| < \infty$, it follows that $W < \infty$, and moreover, $|W_n| \le W$. Hence,

$$
\begin{aligned}
M_{n,2}^2 &= \left\| \int_{A_n^c} f(x) \log f(x) \, dx \right\|_2^2 \\
&= \left\| \int_{\mathbb{R}} f(x) \log f(x) \, \mathcal{I}(x \in A_n^c) \, dx \right\|_2^2 \\
&= \mathbb{E}\,[W_n],
\end{aligned}
\tag{16}
$$

and the Lebesgue Dominated Convergence Theorem guarantees that

$$\lim_{n \to \infty} M_{n,2}^2 = \lim_{n \to \infty} \mathbb{E}\,[W_n] = \mathbb{E}\left[\lim_{n \to \infty} W_n\right] = \mathbb{E}\,[0] = 0,$$

thereby proving the corollary.  □

**Author Contributions:** Conceptualization, T.F. and H.S.; Methodology, T.F. and H.S.; Formal Analysis, T.F. and H.S.; Investigation, T.F. and H.S.; Writing—Original Draft Preparation, T.F.; Writing—Review & Editing, H.S.; Supervision, H.S.; Funding Acquisition, H.S. Both authors have read and agreed to the published version of the manuscript.

**Funding:** This research is supported in part by the Simons Foundation Grant 586789.

**Acknowledgments:** The authors are grateful to the referees and Daniel J. Henderson for carefully reading the paper and for insightful suggestions that significantly improved the presentation of the paper. The research is supported in part by the Simons Foundation Grant 586789 and the College of Liberal Arts Faculty Grants for Research and Creative Achievement at the University of Mississippi.

**Conflicts of Interest:** The authors declare no conflict of interest.

## Appendix A

In the paper Wu et al. (2010), Wu et al. establish results that are very useful in the proof section. Here, we briefly survey their definitions and results which show that the kernel density estimator for one-sided linear processes enjoys properties similar to the independent case—see Stute (1982). Their work identifies conditions under which the kernel density estimator enjoys strong uniform consistency for a wide class of time series. Included is the linear process in (2).

As is common in analysis of time series, we allude to an independent and identically distributed collection $\{\varepsilon_i : i \in \mathbb{Z}\}$ of random variables, typically referred to as the innovations. Note that many time series models fit the form

$$X_n = J(\cdots, \varepsilon_{n-1}, \varepsilon_n),
\tag{A1}$$

which regards the $X_n$ as a system dependent on the innovations. Note here that $J$ is some measurable function which is referred to as the filter. In this context, we also need to define the sigma algebras

$$\mathcal{F}_n = \sigma\{\varepsilon_n, \varepsilon_{n-1}, \cdots\},$$

where $n \in \mathbb{Z}$. In addition, let $\varepsilon_0'$ be an independent and identical copy of $\varepsilon_0$ which is, of course, independent of all the $\varepsilon_i$. For $n \geq 0$, define

$$\mathcal{F}_n^* = \sigma\{\varepsilon_n, \varepsilon_{n-1}, \cdots, \varepsilon_1, \varepsilon_0', \varepsilon_{-1}, \cdots\},$$

and for $n < 0$, put $\mathcal{F}_n^* = \mathcal{F}_n$.

Define the $l$-step ahead conditional distribution by

$$F_l(x|\mathcal{F}_k) = P(X_{l+k} \leq x|\mathcal{F}_k),$$

where $l \in \mathbb{N}$ and $k \in \mathbb{Z}$. When it exists, the $l$-step ahead conditional density is

$$f_l(x|\mathcal{F}_k) = \frac{d}{dx}F_l(x|\mathcal{F}_k).$$

As Wu et al. (2010) note, a sufficient condition for the existence of a marginal density of (A1) is that $f_1(x|\mathcal{F}_0)$ exists and is uniformly bounded almost surely by some $M \in \mathbb{R}^+$. We shall refer to this as the marginal condition. Similarly, $F_l(x|\mathcal{F}_k^*) = P(X_{l+k}^* \leq x|\mathcal{F}_k^*)$, where $X_{l+k}^* = X_{l+k} - a_{l+k}\varepsilon_0 + a_{l+k}\varepsilon_0'$ if $l + k \geq 0$ and $X_{l+k}^* = X_{l+k}$ if $l + k < 0$. Also, $f_l(x|\mathcal{F}^*) = \frac{d}{dx}F_l(x|\mathcal{F}_k^*)$.

With this setup, the authors introduce the following measures of the dependence present in the system (A1). Now, for $k \geq 0$, define a pointwise measure of difference by

$$\theta_k(x) = \|f_{1+k}(x|\mathcal{F}_0) - f_{1+k}(x|\mathcal{F}_0^*)\|_2$$

and an $\mathcal{L}^2$-integral measure of difference over $\mathbb{R}$ by

$$\theta(k) = \left[\int_{\mathbb{R}} \theta_k^2(x)\, dx\right]^{\frac{1}{2}}.$$

Finally, define an overall measure of difference by

$$\Theta(n) = \sum_{j \in \mathbb{Z}}\left(\sum_{k=1-j}^{n-j} |\theta(k)|\right)^2.$$

The distances on the derivatives are defined similarly, as given below.

$$\psi_k(x) = \|f_{1+k}'(x|\mathcal{F}_0) - f_{1+k}'(x|\mathcal{F}_0^*)\|_2,$$

$$\psi(k) = \left[\int_{\mathbb{R}} \psi_k^2(x)\, dx\right]^{\frac{1}{2}}, \quad \text{and}$$

$$\Psi(n) = \sum_{j \in \mathbb{Z}}\left(\sum_{k=1-j}^{n-j} |\psi(k)|\right)^2.$$

With this setup, we can now report the following result of (Wu et al. 2010, Theorem 2).

**Theorem A1.** *Assume that, for some positive $r$ and $s$, we have that $K \in \mathcal{C}^r$ is a bounded function with bounded support and that $X_n \in \mathcal{L}^s$. Further, assume the marginal condition, and assume that $\Theta(n) + \Psi(n) = O(n^\alpha \tilde{l}(n))$, where $\alpha \geq 1$ and where $\tilde{l}$ is a slowly varying function. If $\log n = o(nh_n)$, then*

$$\sup_{x \in \mathbb{R}} |f_n(x) - \mathbb{E}f_n(x)| = O\left(\sqrt{\frac{\log n}{nh_n}} + n^{-\frac{1}{2}}l(n)\right),$$

*where $l(n)$ is another slowly varying function.*

Now consider our particular case when the filter is the linear process of (2). In view of our assumption that the innovations have finite variance and because we assume the coefficients are square-summable, $X_n \in \text{Ł}^2$. Moreover, we assume all of the bandwidth, kernel, and density conditions listed earlier, from which it easily follows that the marginal condition is satisfied. For the short memory linear process (under the aforementioned assumptions), Wu et al. (2010) demonstrated that $\Theta(n) + \Psi(n) = O(n)$. Also, notice that condition **B.1** implies that $\log n = o(nh_n)$. Therefore, the theorem of Wu et al. (2010) applies to (2).

In addition, the well-known Taylor series argument under the conditions **K.2** and **K.3**, as well as **D.3**, yields

$$\sup_{x \in \mathbb{R}} \left| \mathbb{E}\left[f_n(x)\right] - f(x) \right| = O(h_n^2),$$

so, collectively, we see that

$$\sup_{x \in \mathbb{R}} \left| f_n(x) - f(x) \right| = O\left( \sqrt{\frac{\log n}{nh_n}} + n^{-\frac{1}{2}} \, l(n) + h_n^2 \right).$$

Basic methods of differential calculus show that $\sqrt{\frac{\log n}{nh_n}} + h_n^2$ is minimized when $h_n$ satisfies **B.1**. Indeed, the optimum value of $h_n$ has the exact order of $\left( \frac{\log n}{n} \right)^{\frac{1}{5}}$.

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
