# Peer review of "Shannon Entropy Estimation for Linear Processes"

_jrfm, doi:10.3390/jrfm13090205_

Round 1
Reviewer 1 Report
The authors consider an estimation of the Shannon entropy and show the convergence almost surely.
Recommendation: The paper is acceptable provided it is subjected to the following major revision based on the following comments.
Comments to Authors
1. The authors should consider a recent methods of bandwidth selection in order to be able to consider the KDE estimators. For this purpose, the authors should update the references of kernel density estimation, and include the review of the main and recent works on bandwidth selection (see for instance):
Duin, R. P. W. (1976). On the choice of smoothing parameters of Parzen estimators of Probability density function. IEEE Trans. Comput. C-25, 1175–1179.
Rudemo, M. (1982). Empirical choice of histograms and kernel density estimators. Scand. J.
Stat. 9, 65–78.
Slaoui, Y. (2014). Bandwidth selection for recursive kernel density estimators defined by stochastic approximation method. J. Probab. Stat 2014, ID 739640, doi:10.1155/2014/739640.
Slaoui, Y. (2018). Bias reduction in kernel density estimation. Journal of Nonparametric Statistics, 30, 505–522.
2. I think above references need to be addressed.
3. The authors can added some discussion and perspective/open questions linked to the considered problem.
4. The manuscript needs to be carefully proofread by the authors.
Minor comments In the abstract:
1. ”almost surely and in” should be ”almost surely in”
2. The authors should check all the typos of the paper, many are in the reference list.

Reviewer 2 Report
The result is interesting. I checked for correctness, and the paper is well written.
Slight revisions recommended:
But I recommend that 1) the authors include some details regarding particular engineering applications. Also 2) that they add comments comparing their own results with earlier publications in the general area.
Round 2
Reviewer 1 Report
No more comments